# A New Therapeutic Application of Platelet-Rich Plasma to Chronic Breast Wounds: A Prospective Observational Study

**DOI:** 10.3390/jcm9103063

**Published:** 2020-09-23

**Authors:** Juan de Dios Berná-Serna, Florentina Guzmán-Aroca, José A. García-Vidal, Dolores Hernández-Gómez, Ana Azahara García-Ortega, Tomás Chivato Martín-Falquina, Antonio Piñero-Madrona, Juan de Dios Berná-Mestre

**Affiliations:** 1Department of Radiology, Hospital Clínico Universitario “Virgen de la Arrixaca”, Ctra. Madrid-Cartagena, 30120 El Palmar (Murcia), Spain; jdberna@gmail.com ( J.d.D.B.-S.); lhgomez00@hotmail.com (D.H.-G.); anaazaharagarcia@gmail.com (A.A.G.-O.); mesjubermu@hotmail.com (J.d.D.B.-M.); 2Instituto Murciano de Investigación Biosanitaria “Virgen de la Arrixaca” (IMIB-Arrixaca), 30100 Murcia, Spain; garciavidal@um.es (J.A.G.-V.); pineromadrona@gmail.com (A.P.-M.); 3Departament of Physiotherapy, University of Murcia, Campus de Espinardo, 30100 Murcia, Spain; 4Radiopharmacy Unit. Hospital Clínico Universitario “Virgen de la Arrixaca”, Ctra. Madrid-Cartagena, 30120 El Palmar (Murcia), Spain; Tomas.chivato@carm.es; 5Department of Surgery, Hospital Clínico Universitario “Virgen de la Arrixaca” and Universidad de Murcia, Murcia, Spain

**Keywords:** breast, platelet-rich plasma, chronic wound, breast conservation therapy

## Abstract

The aim of this study was to investigate the usefulness of platelet-rich plasma (PRP) treatment for chronic wounds (CWs) of the breast. A prospective study was performed in 23 patients with CW of the breast who were treated with PRP. The procedure was repeated until the wound was closed completely. The study included patients with a history of breast cancer (*n* = 8) and patients without cancer (*n* = 15). The treatment with PRP was successful in all cases and observed in ≤4 weeks in 82.6% (19/23) of patients. The patients without breast cancer showed significantly less time for wound closure than the patients with a history of breast cancer. Moreover, a greater number of PRP treatments were necessary to achieve wound closure in patients undergoing conservative breast treatment. No patients had complications associated with the application of PRP. Conclusions: To the best of our knowledge, this is the first study to reveal that PRP treatment for CWs of the breast is safe, simple, useful and well-tolerated by patients.

## 1. Introduction

Chronic wounds (CWs) are a major health problem and have a huge impact on a personal, professional and socioeconomic level [1,2,3]. It is estimated that between 1% and 2% of the population in developed countries will suffer from a CW during their lifetime [4]. A CW is considered when the healing process is delayed for a period of ≥6 weeks [3,5]. CWs are a challenge for professionals when they do not respond to conventional treatment. New therapeutic strategies have been used in recent years such as platelet-rich plasma (PRP) for CW healing [6,7,8,9].

PRP is defined as autologous blood with a higher concentration of platelets than in peripheral blood [6,10]. PRP contains cytokines, chemokines, growth factors and a fibrin scaffold, which play a major role in wound healing and tissue regeneration [6,11,12]. PRP is also widely used in different areas of medicine such as aesthetic and regenerative medicine [13], oral and maxillofacial surgery [14], orthopedic surgery and sports medicine [15], musculoskeletal procedures [16] and gynecology [17].

Several studies [8,9,18,19] demonstrate the usefulness of PRP application to CWs. The combination of fat grafting and PRP also has significant potential to improve wound healing [20]. It has likewise been proposed that allogeneic PRP is an effective and safe adjuvant treatment for chronic wounds [21]. We consider that the percentage of reduction in the initial wound area after application of PRP, especially in the first two weeks, might be a good prognostic indicator of wound healing.

To the best of our knowledge there are no studies in the literature on the therapeutic application of PRP for CWs of the breast. We consider PRP to be a therapeutic option, and the aim of the present study is to investigate the effectiveness of PRP treatment for CWs of the breast.

## 2. Patients and Methods

### 2.1. Patients

This prospective observational study was conducted in patients with CWs of the breast (≥8 weeks) who received treatment with PRP between March 2013 and December 2019. The patients were referred to us from our hospital’s breast unit. Prior to the application of PRP treatment, the patients had a complete laboratory test to evaluate cell count and serology tests to detect infectious agents, including diseases such as syphilis, hepatitis B, hepatitis C and HIV I/II. The criteria for inclusion were as follows: age >18 years; CW of the breast (≥8 weeks); patients with a clinical history of breast cancer (*n* = 8) receiving conservative treatment (lumpectomy) with tumors of ≤3 cm; patients with benign breast lesions (*n* = 13) undergoing surgical excision and patients who had breast reduction (*n* = 2). The exclusion criteria were pregnancy, multicentric/multifocal breast cancer or tumors of >3 cm, coagulopathies and patients with syphilis, hepatitis B, hepatitis C and HIV. The study was conducted according to the guidelines of the Helsinki Declaration and written approval was obtained by our Institutional Ethics Committee (“Virgen de la Arrixaca” University Hospital IRB Reference number: 2020–06-03–HCUVA). Written informed consent to participation in the study, and to data collection, was obtained from the patients prior to the start of the study. 

### 2.2. PRP Preparation

Depending on the size of the wound we extracted, between 40 and 60 mL of peripheral blood with two syringes (S1 and S2) containing anticoagulant A (mixture of 1.49% citric acid and 2.5% sodium citrate) and B (3.8% sodium citrate), respectively. The S2 blood was centrifuged at 1000 g (2500 rpm) for 5 min to yield cell-free plasma. Then, the S1 blood was centrifuged at 180 g (1050 rpm) for 15 min for the blood cells and leukocytes to settle. The supernatant was centrifuged again at 640 g (2000 rpm) for 10 min. The resulting platelet button was resuspended in the desired volume of cell-free plasma to obtain PRP. To activate the liberation of platelet growth factors, we added 5% of the volume to a calcium chloride solution (0.91 N). The entire procedure was carried out in a class A safety cabinet. We finally obtained between 3 and 6 mL of PRP, which was used in both gel and liquid forms.

### 2.3. Treatment Protocol

Assessment was done by an interventional radiologist using ultrasound (US) to measure the depth of the wound. The size of the wound (length × width) was measured by a graduated ruler and photographed with a digital camera. The depth of the wound was determined by ultrasound (US) examination using an Acuson S2000 ultrasound system (Siemens, Erlangen, Germany) equipped with an 18L6HD transducer, or Philips EPIQ 7 with an 18–5L MHz (Philips Healthcare, Bothell, WA, USA). Before PRP application, the wound was cleaned with sterile saline solution and betadine and the edges were debrided with a scalpel, where necessary, after administration of local anesthesia (2% mepivacaine) to the contour of the wound. PRP application to the wounds was a combination of gel and liquid. First of all, a 14G Abbocath was used under US guidance to inject PRP liquid percutaneously into wounds of ≥1 cm in depth or directly into the bed of the wound. PRP gel was then applied to the bed of the wound (Figure 1). Following this, the wound was covered with a collagen dressing soaked in PRP liquid, which was further covered with sterile gauzes and adhesive dressings. The patients were recommended relative rest, to avoid pressure on the wound and to take anti-inflammatories. The patients were reviewed at 5–7 days (week 1: W1), when the dressings were removed with the utmost care, using saline solution, and the wound was re-evaluated. In cases in which the wound had not closed completely, it was cleaned with saline solution and covered with a dressing soaked in saline solution. In these cases, PRP was applied at an interval of two weeks from the initial treatment.

The wound was considered healed when complete closure was observed, our definition of wound healing being full epithelialization and no drainage without the need for additional dressing [22]. Clinical follow-up and US were performed weekly from W1 to W7, between six and nine months and in cases of patients with conservative treatment for breast cancer annually with US and mammography.

### 2.4. Data Collection

The information recorded for each patient included age, duration of symptoms, height, weight, body mass index (BMI), smoking, diabetes, duration time of the wounds and their location in the breast. Therapeutic data were also collected such as that of the patients undergoing breast-conservation therapy (BCT), which consists of breast-conserving surgery (lumpectomy) and postoperative radiotherapy, as well as the surgical procedures performed in the cases of benign breast lesions, breast reduction, surgical abscess drainage and postsurgery complications. The following were regarded as postoperative complications: infected seroma, wound dehiscence and fat necrosis. We also determined the depth of the wound with US, and the US appearance of the wound after healing. The area of the wound was calculated using the formula Length × width × 0.785 [23]. The percentage area reduction (PAR) was calculated with the formula PAR for the index wound: PAR = [(AI–AW)/AI)] × 100, where AI is the initial area and AW is the area at weeks after treatment [24].

### 2.5. Statistical Analysis

Descriptive statistics were used to characterize the cohort at baseline and to describe the number of sessions until the wound closed. Cox’s proportional hazards regression univariate and multivariate analysis was used for estimating the hazard ratios (HRs) and 95% confidence intervals (CIs) for the time to healing. The groups defined by meaningful and categorical variables (*p* < 0.05) from the multivariate analyses were compared to identify significant differences. Normally distributed continuous variables were compared using the independent samples t-test, and variables that were not normally distributed were compared with the Mann-Whitney U test. Categorical variables were analyzed using the chi-squared test or Fisher’s exact test. Moreover, Kaplan Meier analysis and the log rank test were used to test for significant differences between the survival curves of the variables influencing the time to curation and to compare survival probabilities in order to identify significant clinical differences between the groups.

## 3. Results

### 3.1. Participants and Wound Healing 

The mean age ± SD of the 23 patients undergoing PRP (Figure 2) was 49.6 ± 16.0 years (range: 31–81 years). Table 1 shows the characteristics of the patients in this study. The mean duration of the conventional treatment of the wounds was 10.8 ± 2.6 weeks (range: 8–16 weeks). The initial mean area of the wounds was 304.8 ± 230.0 mm^2^ (range: 40–686 mm^2^) and the mean depth of the wound was 16.2 ± 3.4 mm (range: 9–27 mm). Breast-conservation therapy (BCT) was administered to eight patients in the present study, of whom three were unable to receive radiation therapy followed by lumpectomy due to wound dehiscence (*n* = 2) and seroma (*n* = 1); the wound was closed at 2 and 4 weeks (dehiscence) and 4 weeks (seroma), after PRP treatment. However, in these patients the time to resolution of the wound, considering the duration of conventional treatment, was 10 and 12 weeks (dehiscence) and 15 weeks (seroma). In patients with BCT the mean length of follow-up was 60.7 ± 9.3 months (range: 51–71 months).

### 3.2. Characteristics Influencing the Time to Wound Healing

Table 2 shows the hazard ratio and *p*-value of each characteristic estimated by the analyses using the Cox proportional hazard regression models. The statistical interaction between the variables’ origin and initial area (HR = 1.00 (0.99–1.01; *p* = 0.402) showed that the effect of the area on the time to healing did not differ between our patients overall. The survival curves in Figure 3 reflect the effects of the wounds closed between patients with a history of breast cancer and without breast cancer measured at baseline. The graph shows that the patients without breast cancer had significantly less time to wound closure than the patients with a history of breast cancer (X2(1) = 8.34; *p* = 0.004). Furthermore, Figure 4 shows that the effect appeared to differ according to sizes >370 mm, as the patients with BCT required a longer number of weeks.

The wounds were healed in all cases in the present study. A mean of 1.91 ± 0.79 (range: 1–4) treatments of PRP were applied to each patient, although the patients with BCT had mean PRP treatments of 2.37 ± 0.85 (range: 1–4), whereas the women without breast cancer had a mean of 1.66 ± 0.59 (range: 1–3). Figure 5 and Figure 6 show examples of wound treatment with PRP. No patients had complications associated with the application of PRP during or after the treatment.

## 4. Discussion

To our knowledge, this is the first study that applied PRP therapy in CWs of the breast. The findings obtained show that the procedure was successful in all cases. PRP has become recognized in recent years as an emergent treatment for chronic wounds, especially when conventional treatments fail [3,7,8,9].

CWs are a common pathology and represent a major health problem. They affect the patients′ quality of life and involve a considerable financial cost [1]. There is still uncertainty regarding the effectiveness of PRP to cure CWs. Several studies [8,9,18,19] show good results with PRP treatment in CWs. However, it is unclear whether PRP influences the healing of chronic wounds, as the existing evidence is sparse and of low quality [25]. There is also a considerable methodological variability and lack of standardization for the use of PRP. It is, therefore, essential to establish standardized clinical protocols for PRP application to CWs.

The present study achieved excellent results when applying a standardized PRP protocol for the treatment of CWs of the breast. Healing of the wound was seen in all the patients, and in 82.6% (19/23) this was ≤4 weeks. Healing of the wound was quicker in the patients with no history of breast cancer. Two patients undergoing BCT who presented with a wound in the inferoexternal quadrant close to the inframammary groove, received four and five PRP treatments, respectively, to heal the wound. On the basis of our results, we believe that a good prognostic indicator for wound healing time is the percentage of reduction in the initial wound area. The mean initial area of the wound was not a good prognostic indicator for healing. In the present study, the mean initial wound area was less in the cancer patients than in the noncancer patients. However, the cancer patients required more PRP treatments than those without cancer, and their wound healing time was longer.

BCT is the standard option for women with early-stage breast cancer, and postlumpectomy radiotherapy is still a key component to achieve a 50–66% reduction in local recurrence [26,27]. An interval of six to eight weeks has been suggested from the time of tumor resection to the start of radiotherapy [28,29]. Three cases in the present study with postsurgical complications had wound healing at 10 to 15 weeks. However, wound healing with PRP application was observed at two to four weeks. We consider that to avoid delays in administering the radiotherapy, PRP could be applied to postsurgical wounds and from four weeks onwards if they are not resolved with conventional treatment. It is also very important to start radiotherapy as soon as possible to minimize the risk of local recurrence.

Given the autologous nature of PRP, it is considered a safe product. There is also no evidence to support the oncogenic potential of PRP. In an experimental study [30], the findings observed indicated that PRP can be safely used in short and long-term administration without concern for tumorigenesis. In the present study PRP was applied to eight patients with a history of breast cancer diagnosed in an early stage with tumors of ≤3 cm and disease-free margins. It is also important to stress that in >4 years of follow-up, no malignant breast pathology was detected in these patients.

There were several limitations to the present study. The study population was relatively small, and the data were obtained retrospectively. Moreover, to avoid interobserver variability, the application of PRP to the wounds was done by a radiologist with experience in breast interventionism, who also measured the depth of the wounds with US and determined the area of the wounds. We believe further studies are necessary to optimize the PRP protocol, especially in wounds with initial areas of ≥370 mm^2^, as PRP treatment could be applied weekly with a view to reducing the time of wound healing.

## 5. Conclusions

To the best of our knowledge, this is the first study in which PRP treatment has been used for the management of CWs of the breast. The results obtained show that the procedure is useful, safe, simple and well-tolerated by patients. All the wounds were healed by treatment with PRP. Furthermore, we recommend using PRP in postlumpectomy wounds to avoid delays in administering radiotherapy to patients with an early diagnosis of breast cancer.

## Figures and Tables

**Figure 1 jcm-09-03063-f001:**
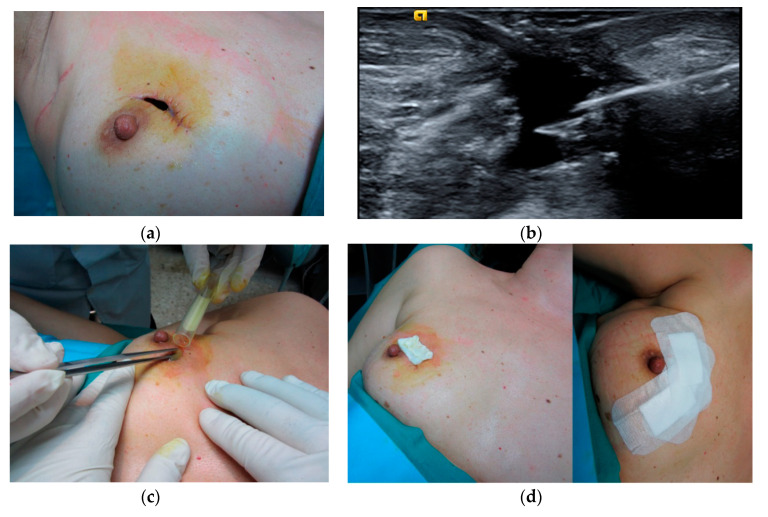
Application of platelet-rich plasma (PRP) in a case of wound dehiscence. (**a**) Pretreatment, (**b**) ultrasound image showing injection of PRP in the wound cavity, (**c**) PRP gel applied to the wound, (**d**) the wound was covered with a collagen dressing and adhesive dressings.

**Figure 2 jcm-09-03063-f002:**
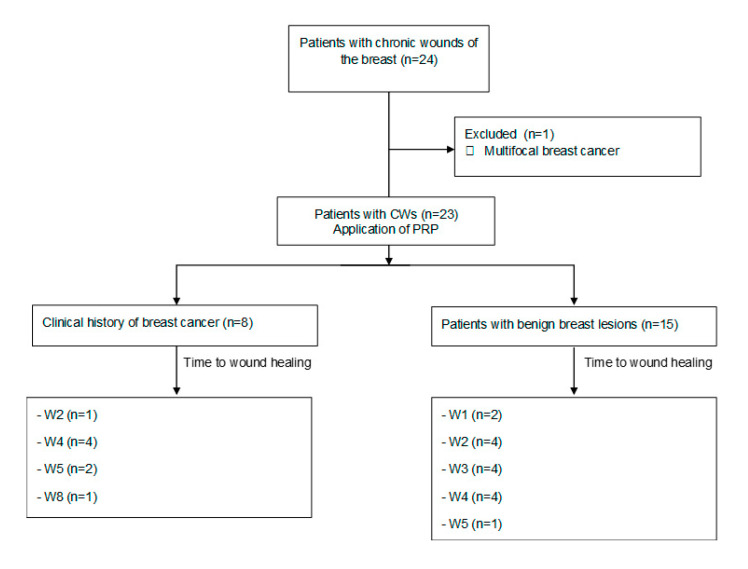
Flow diagram of the patients in this study. CWs: chronic wounds; PRP: platelet-rich plasma; W: week.

**Figure 3 jcm-09-03063-f003:**
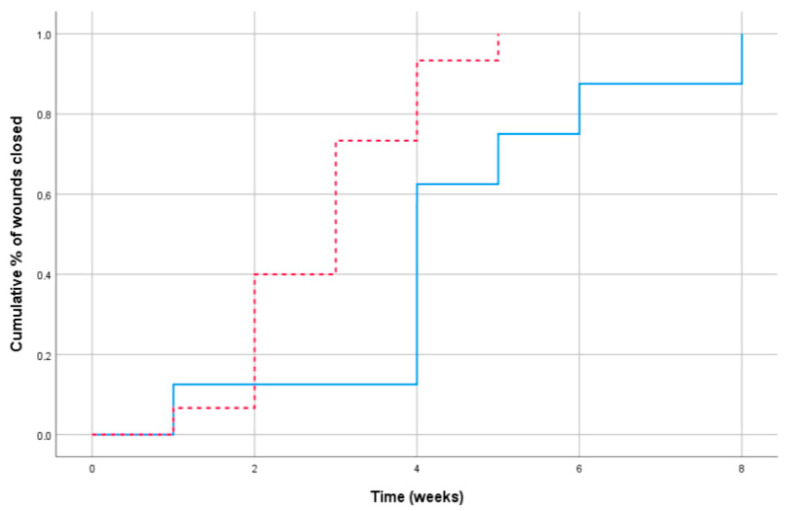
Cumulative percentage of wound closure by weeks: patients without breast cancer (dashed line) and patients with conservative treatment for breast cancer (solid line).

**Figure 4 jcm-09-03063-f004:**
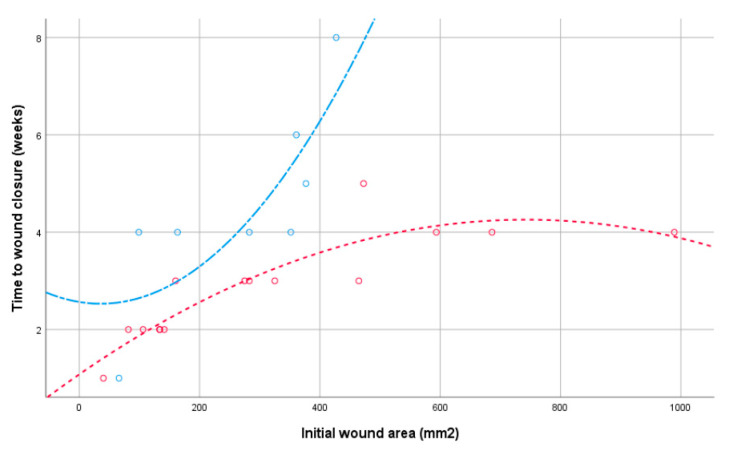
Distribution of wound healing times by initial wound area: patients without breast cancer (red line) and patients with conservative treatment for breast cancer (blue line).

**Figure 5 jcm-09-03063-f005:**
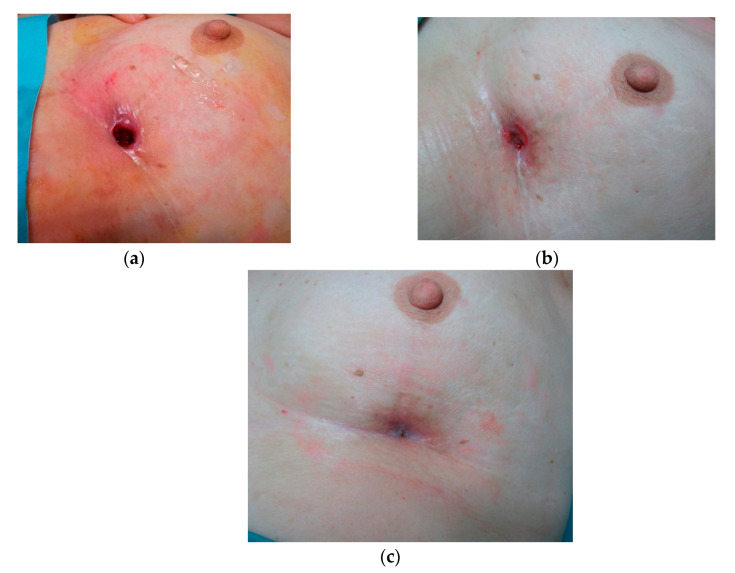
Patient receiving conservative treatment for breast cancer who required 4 PRP treatments for the wound to heal. (**a**) pretreatment, (**b**) 6 weeks, (**c**) wound closure.

**Figure 6 jcm-09-03063-f006:**
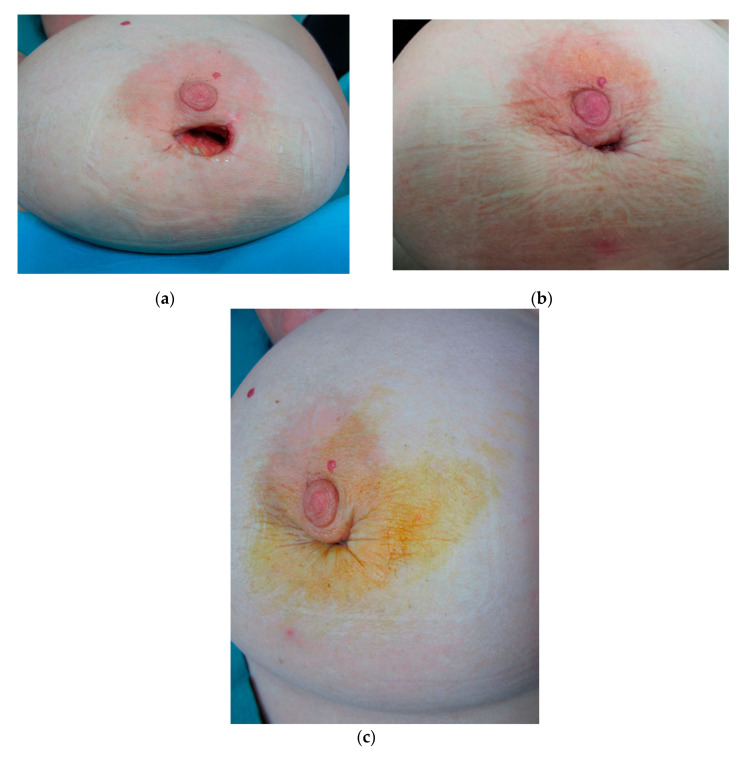
Patient undergoing surgical excision for a benign breast lesion who required two PRP treatments for the wound to heal. (**a**) pretreatment, (**b**) three weeks, (**c**) wound closure.

**Table 1 jcm-09-03063-t001:** Characteristics of all participants and patients with a history of breast cancer and patients without breast cancer.

Demographics and Background	All Samples (*n* = 23)	With Cancer (*n* = 8)	Without Cancer (*n* = 15)	*p*-Value
Age (years), mean ± SD	49.6 ± 16.0	62.2 ± 15.6	42.8 ± 11.8	0.003
BMI *, mean ± SD	27.2 ± 2.7	27.8 ± 2.9	26.9 ± 2.6	0.497
Smoker	11 (47.8)	0 (0)	11 (73.3)	0.001
Without Diabetes	21 (91.3)	7 (87.5)	14 (93.3)	0.999
Clinical characteristics				
Duration of previous treatments (weeks), mean ± SD	10.8 ± 2.6	12.0 ± 2.7	10.1 ± 2.4	0.106
Abscess with drainage	4 (17.4)	1 (12.5)	3 (20)	1.0
Seromas with drainage	4 (17.4)	4 (50.0)	0 (0)	0.008
With dehiscence	3 (13.0)	2 (25.0)	1 (6.7)	0.269
With fat necrosis	3 (13.0)	1 (12.5)	2 (13.3)	0.999
Initial wound area (mm^2^), Mean ± SD	304.8 ± 230.0	265.9 ± 138.0	325.6 ± 268.8	0.565
Right breast	14 (60.9)	5 (62.5)	9 (60.0)	1.0
Wound location				0.123
Upper outer quadrant	6 (26.1)	2 (25.0)	4 (26.7)	
Periareolar	15 (65.2)	4(50.0)	11 (73.3)	
Lower outer quadrant	2 (8.7)	2 (25.0)	0 (0)	

Abbreviations: * Body mass index.

**Table 2 jcm-09-03063-t002:** Predictors of time of wound closure in Cox’s proportional hazards models.

Variables	Univariate Models HRs * (95% CI) *p*-Value	Multivariate Model
Body mass index	0.93 (0.82–1.06) 0.305	
Smoker	2.21 (0.87–5.58) 0.101	
Nondiabetic	1.67 (0.38–7.05) 0.516	
Weeks in previous treatment	0.84 (0.69–1.01) 0.067	
With cancer (vs. without)	2.72 (1.02–7.29) 0.046	7.08 (2.01–24.92) 0.002
Had abscess drainage	1.28 (0.43–3.87) 0.659	
Had fat necrosis	1.10 (0.32–3.86) 0.883	
Right breast	2.33 (0.84–6.47) 0.110	
Location		
Upper outer quadrant	Reference	
Periareolar	0.97 (0.37–2.53) 0.969	
Lower outer quadrant	N.A. ** 0.957	
Initial area (mm^2^)	0.99 (0.99–1.00) 0.050	0.99 (0.992–0.998) 0.004

Abbreviations: * Hazard ratio. ** not applicable.

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
