# Peer review of "A New Therapeutic Application of Platelet-Rich Plasma to Chronic Breast Wounds: A Prospective Observational Study"

_jcm, 2020, doi:10.3390/jcm9103063_

Round 1
Reviewer 1 Report
Dear Authors,
Thank you for the opportunity to review your manuscript describing the use of PRP in breast wound complications. This is certainly of interest only from a scientific perspective but also clinically. Therapies that propose to improve wound healing outcomes are worthy of rigorous investigation. Your manuscript provides baseline data for further work. There are several comments for the author team to address below;
- Study Methodology; from a methodological standpoint, this study reads as a prospective observational series, not a retrospective study. Therefore, the correct methodological term is prospective observational series. As this study has not engaged a randomised control trial methodology please remove the word ‘effectiveness’ or ‘effective’ from the manuscript as this is reserved for large rigours powered and controlled trials. Perhaps the more appropriate term is ‘observed trend/s’, ‘observed pattern’. Please also remove retrospective study from text as you have conducted this study using a prospective observation method. i.e. prospectively recruited and consented participants, applied an intervention to a series of participants (no controls), and made observations from baseline statistics.
- Change manuscript title: The more appropriate title for your study would be; ‘A new therapeutic application of platelet-rich plasma 3 to chronic breast wounds: a prospective observational study’. This is indicative of the study you have conducted and informs the reader of the methodology you have used.
- Background: Line 43. Please provide a critical appraisal of current studies in the area as part of your background section. Please describe and discuss the gaps in current research and how your study proposes to address these gaps. There are a number of review papers that are also missing, please see the contemporary work of Liao et al, Smith et al 2019 etc.
- Line 60; please describe what the consent was sought for. For example; written informed consent was obtained from the patient prior to study commencement, for participation in the study and data collection.
- Study design and major limitation: This study has made observations in participants with a cancer diagnosis and healthy non-cancer participants. Your conclusions in relation to the impact of the use of the PRP gel in terms of increasing healing times are unstable as you are comparing two different populations, one of which is already predisposed to delayed wound healing due to a cancer diagnosis and treatment. Authors, please provide a detailed and rigorous discussion in regards to this major study limitation and the impact on conclusions you or the reader may draw. Generalisations to the wider population are considerably limited in this study. In Tables 1 & 2 you have compared cancer vs non cancer outcomes in breast complications using the intervention. It is best to compare like with like, so comparing outcomes in cancer patients who received or did not receive the gel, has more meaning due to the biological and clinical implications. Table 2 shows that you have accelerated healing outcomes in the non-cancer patients, one may anticipate this due to the healthy participants compared to cancer participants. The question is does this table add value to your manuscript? Are you able to construct a table showing results of cancer patients only comparing those who received the gel and those that did not? If you were unable to source this data, this would certainly be your next study and will be far more telling.
- Please provide a CONSORT flow chart for your study clearly outlining screening numbers, inclusion/exclusion criteria & numbers and final number of participants. See the following link to assist: https://www.equator-network.org/wp-content/uploads/2015/10/STROBE_checklist_v4_combined.pdf and for CONSORT; https://www.equator-network.org/reporting-guidelines/consort/. As you have not randomised, omit and just list numbers recruited.
- Please provide information on the clinical setting from where you recruited your participants (under patients and methods).
- Suggested change line 74 to; assessment was conducted by an interventional radiologist using ultrasound to determine/measure……...
- Line 92, please explain how you determined complete closure,
i.e the wound was assessed by a clinician/s who used a wound care plan assessment to measure and…
- Tables 1&2 please provide a legend/key for the coloured lines inside the table, not as subscript to the table.
- Please describe why you have conducted a logistic regression in the methods as this seems out of context in the overall manuscript. Whilst you have determined a number of predictors, which is why one uses this test, the conclusion you have drawn (lines 134 &135) is confusing. Please consult a statistician and rewrite this section. Furthermore, please review characteristics influencing wound healing in the background section, as section 3.2 appears out of context.
12. Please discuss the implications of why those patients with cancer required more treatments than those without a cancer diagnosis
Thank you again for the opportunity to review your manuscript. Look forward to receiving the revised manuscript in due course.
Best
Reviewer
Author Response
Please see attachment, thank you.

Reviewer 2 Report
In the presented retrospective study “A new therapeutic application of platelet-rich plasma to chronic breast wounds” the authors show that PRP treatment is a safe way to treat chronic wounds (CWs) of the breast. Patients suffering from CWs after conservative treatment (lumpectomy) of tumors or surgical excision of benign breast lesions for several weeks were healed due to the PRP therapy.
The study is well conducted, the results are clearly presented and the conclusions are justified. However, the results of the study are not very surprising, and the main result – that the wound closure is faster for smaller initial wound areas and for patients without breast cancer – has no impact on the clinical therapy decision. The study could have benefitted from a comparison to a group of patients that did not receive a PRP therapy. Nonetheless, the data show the safety and the effectiveness of autologous PRP treatment and will be helpful for patients with CWs of the breast.
Only a few minor concerns should be addressed:
-Maybe the Y-axis in Figure 2 should be labeled up to 100, not to 1?
-In Figure 3, the blue line and red line should be indicated in the figure legend, too (not only in figure 2)
-The authors should discuss the results from table 2 in more detail (even if the p-value indicates no significance, the results for some of the variables should be discussed
-the initial wound area varies greatly and might be the main factor causing the closure time. Maybe some of the variables in table 2 show significant impacts if only patients with a comparable initial wound area are compared?
Author Response
Please see the attachment, thank you

Round 2
Reviewer 1 Report
Dear authors,
Thank you, no further comments. Good luck with your ongoing research.
Author Response
Many thanks for your comments and observations which have permitted us to improve the manuscript.
- The type of study from retrospective to prospective was changed in the title and in the abstract but not in Patients and methods (lines 55-56)
We have changed from retrospective to prospective in Patients and methods
- Treatment protocol lines 81-84: The sentence “Assessment was done by an interventional radiologist using ultrasound (US) to measure.” and “The size of the wound (length x width) was measured by a graduated ruler and 83 photographed with a digital camera.” are not clear and conflicting. How was the wound measurement actually made? By US or visually with a ruler? Please clarify
The wound measurement was done using ultrasound to measure the depth of the wound (we have added it in the manuscript) and the size of the wound (length x width) was measured by a graduated ruler